# A Splice Variant in *SLC16A8* Gene Leads to Lactate Transport Deficit in Human iPS Cell-Derived Retinal Pigment Epithelial Cells

**DOI:** 10.3390/cells10010179

**Published:** 2021-01-18

**Authors:** Laurence Klipfel, Marie Cordonnier, Léa Thiébault, Emmanuelle Clérin, Frédéric Blond, Géraldine Millet-Puel, Saddek Mohand-Saïd, Olivier Goureau, José-Alain Sahel, Emeline F. Nandrot, Thierry Léveillard

**Affiliations:** 1Institut de la Vision, Sorbonne Université, INSERM, CNRS, 17 rue Moreau, F-75012 Paris, France; laurence.klipfel@inserm.fr (L.K.); marie.cordonnier@hotmail.fr (M.C.); lea.thiebault@inserm.fr (L.T.); emmanuelle.clerin@inserm.fr (E.C.); frederic.blond@inserm.fr (F.B.); geraldine.millet-puel@inserm.fr (G.M.-P.); olivier.goureau@inserm.fr (O.G.); j.sahel@gmail.com (J.-A.S.); emeline.nandrot@inserm.fr (E.F.N.); 2CHNO des Quinze-Vingts, INSERM-DGOS CIC 1423, 28 rue de Charenton, F-75012 Paris, France; saddekms@gmail.com

**Keywords:** retinal pigment epithelium, lactate transport, induced pluripotent stem cells, splicing, MCT3, age-related macular degeneration

## Abstract

Age-related macular degeneration (AMD) is a blinding disease for which most of the patients remain untreatable. Since the disease affects the macula at the center of the retina, a structure specific to the primate lineage, rodent models to study the pathophysiology of AMD and to develop therapies are very limited. Consequently, our understanding relies mostly on genetic studies highlighting risk alleles at many loci. We are studying the possible implication of a metabolic imbalance associated with risk alleles within the *SLC16A8* gene that encodes for a retinal pigment epithelium (RPE)-specific lactate transporter MCT3 and its consequences for vision. As a first approach, we report here the deficit in transepithelial lactate transport of a rare *SLC16A8* allele identified during a genome-wide association study. We produced induced pluripotent stem cells (iPSCs) from the unique patient in our cohort that carries two copies of this allele. After in vitro differentiation of the iPSCs into RPE cells and their characterization, we demonstrate that the rare allele results in the retention of intron 2 of the *SLC16A8* gene leading to the absence of MCT3 protein. We show using a biochemical assay that these cells have a deficit in transepithelial lactate transport.

## 1. Introduction

Outer retinal degenerations are blinding diseases characterized by the loss of photoreceptors. Age-related macular degeneration (AMD) is the principal cause of irreversible visual impairment in the elderly in industrialized countries. It is a late-onset disease resulting from the interplay between multiple susceptibility genes and environmental factors. AMD primarily affects central vision that is dependent on cones, enriched in the macula at the center of the retina. One of the prominent accepted hypotheses proposes that degeneration results from an initial defect in photoreceptor outer segment processing by the retinal pigment epithelium (RPE) [1]. With aging, the elimination of photoreceptor outer segments becomes increasingly inefficient, leading to progressive accumulation of deposits, known as drusen, at the basal membrane of the RPE [2].

Single nucleotide polymorphisms (SNPs), spread all over the human genome, are used as genetic markers of causative loci since alleles at a given locus are in linkage disequilibrium [3]. By searching for a difference in the frequency of each SNP allele between a population of patients and another population of presumably healthy subjects, genome wide association studies (GWASs) have led to the identification of AMD susceptibility loci. Variants in the complement factor H (*CFH*) gene on chromosome 1q32 have been associated with an increased risk for AMD [4,5,6], implying that the innate immune system plays a significant role in AMD pathogenesis. For *ARMS2/HTRA1*, the second major locus contributing to AMD, it has not yet been possible to determine whether the AMD susceptibility results from the variants in *ARMS2*, the nearby *HTRA1* gene or both [7,8,9,10]. In an effort to identify AMD causative alleles, a recent GWAS identified 52 independently associated common and rare variants distributed across 19 loci [11]. However, translation of these loci into biological insights remains a challenge, as the functional consequences of disease-associated common variants are typically subtle and hard to decipher. Among these loci, the lactate transporter *SLC16A8* gene is the only AMD susceptibility gene that is pointing directly to a dysfunction of photoreceptors. Indeed, the loss of function of photoreceptors in mice with an inactivation of the *Slc16a8* gene demonstrated the essential role of lactate transport for vision [12]. A candidate causal mutation (c.214 + 1G C, rs77968014, odds ratio 1.5) in the immediate 3′ of exon 2, a GT CT, 5′ to the next intron has been identified in a second broader study [13]. The role of lactate transporter MCT3, encoded by the *SLC16A8* gene, in the metabolic ecosystem between photoreceptors and the RPE and the role of cone photoreceptors at the center of the macula suggest that retinal metabolic dysfunction might be at the center of AMD pathogenesis [[14][15],[16]].

Rod-derived cone viability factor (RdCVF), which is produced and secreted by rod photoreceptors, stimulates glucose entry into the cones by the specific interaction between basigin-1 (BSG1) and the glucose transporter SLC2A1 (GLUT1) at the surface of the cones [17]. Glucose, transported into cones, is metabolized by aerobic glycolysis that provides triglycerides used by cones for the renewal of the outer segments [18]. The pyruvate produced by glycolysis can be transported to the mitochondria where oxidative phosphorylation occurs. Pyruvate can alternatively be converted in lactate by the lactate dehydrogenase. Aerobic glycolysis was discovered by Otto Warburg as being specific for cancer cells that prioritize the production of lactate over the production of ATP by oxidative phosphorylation even in the presence of oxygen [19]. Lactate is then transported out of the cones by the lactate transporter MCT1, encoded by the *SLC16A1* gene [20], and through the RPE by two distinct transporters, MCT1 on the apical side towards photoreceptors, and MCT3, encoded by the *SLC16A8* gene, on the basal side, in order to be eliminated in the choroid circulation. Since MCT3 is a facilitated transporter, the direction of the transport is governed by the difference in lactate concentration between the two compartments [21]. The increase in extracellular lactate concentration will reverse the polarity of the transport counteracting aerobic glycolysis flux, leading to the cone outer segment shortening as observed in AMD [22].

The macula is the affected retinal region in AMD, but it is not present in the non-primate species, and consequently a rodent model of AMD is still missing. The identification of an AMD risk allele (rs8135665) and a candidate causal splicing mutation (rs77968014-G) in the *SLC16A8* gene motivated us to produce induced pluripotent stem cells (iPSCs) from skin biopsies of patients homozygous for this allele, and to differentiate them into RPE cells. The functional characterization of these iPSC-derived RPE (iRPE) cells supports the hypothesis that a deregulation of lactate metabolism by the RPE contributes to the pathophysiology of AMD.

## 2. Materials and Methods

### 2.1. Patients

Patients were selected based on their genotype as established by the SNP analysis within the frame of the international age-related macular degeneration genetics consortium (IAMDGC) [13] with the help of a local database, KBaSS [23]. Written informed consent was obtained from the patients, and the study was conducted in accordance with the tenets of the Declaration of Helsinki. All experiments that involved the use of samples obtained from humans were reviewed and approved by French regulatory agencies: CPP Ile de France (2012-A01333–40; P12-02) and the ANSM (B121362-32). Patient phenotype was assessed using visual acuity, fundus photography, optical coherence tomography and autofluorescence imaging (Figure 1). The clinical diagnosis was considered to be adult onset foveal vitelliform dystrophy (AFVD).

### 2.2. Reprogramming of Human Dermal Fibroblasts into Induced Pluripotent Stem Cells

Skin biopsies of patients from the Quinze-Vingts hospital were cut into small pieces and placed into 6-well plates exposed, after 1 h of adhesion to the culture dish, to increasing amounts of “biopsy medium” composed of high glucose Dulbecco’s modified Eagle’s medium (DMEM), supplemented with 10% fetal calf serum (FCS), 2 mM sodium pyruvate, 1% amphotericin B (Fungizone^®^) and 1% penicillin-streptomycin (all from Thermo-Fisher Scientific, Waltham, MA, USA), and placed at 37 °C with 5% CO_2_ in a humidified incubator, as previously described [24]. Half of the medium was changed after 3–5 days upon emergence of dermal fibroblasts, then every week. At confluency, fibroblasts were washed with phosphate buffer saline (PBS), dissociated with TrypLE Select (Thermo-Fisher Scientific) and expanded at a split ratio of 1:3 in “fibroblast medium” composed of high glucose DMEM supplemented with 10% FCS, 1% MEM non-essential amino acids (MEM-NEAA, Thermo-Fisher Scientific) and 1% penicillin-streptomycin. Fibroblasts were frozen at passages 1 and 2 in 1 mL of 90% FCS and 10% dimethyl sulfoxide (DMSO), and stored in liquid nitrogen. Fibroblasts were transduced using the CytoTune^®^-iPS 2.0 Sendai Reprogramming Kit (Thermo-Fisher Scientific) to generate iPSCs under feeder-free conditions onto truncated recombinant human vitronectin (rhVTN-N, Thermo-Fisher Scientific)-coated culture dishes, using the manufacturer’s instructions. Briefly, fibroblasts at passage 2 were plated in 6-well plates at various cell densities in the fibroblast medium and transduced at 50–80% of confluence at the appropriate multiplicity of infection (MOI) of reprogramming vectors (polycistronic *KLF4*-*OCT3/4*-*SOX2*: MOI = 5, *CMYC*: MOI = 5 and *KLF4*: MOI = 3). Medium was replaced the following day, then every other day. At day 7, transduced cells were plated onto vitronectin-coated culture dishes in fibroblast medium, replaced by complete Essential 8™ Medium (Thermo-Fisher Scientific) at day 8. The medium was changed every day and monitored daily for the emergence of iPSC colonies. Emergent undifferentiated iPSC colonies were manually picked under a stereomicroscope according to their human embryonic stem cell (ESC)-like morphology for further subcloning on vitronectin-coated 60-mm diameter organ cell culture dishes (Corning, Oneonta, NY, USA) in Essential 8™ Medium, containing FGF2. Single colony subcloning was performed for at least 5 passages, and at least 10 passages were done before testing for virus free iPSCs and banking. For expansion, iPSC colonies were progressively passaged once a week onto vitronectin-coated 60-mm diameter culture dishes with gentle cell dissociation reagent (Stem Cell Technologies, Grenoble, France) following the manufacturer’s recommendations, and the medium was changed daily. Exogenous reprogramming factors and Sendai virus genome clearance were checked by RT-PCR following the manufacturer’s protocol and primer sets (Appendix A). Absence of mycoplasma contamination was verified using the MycoAlert™ Mycoplasma Detection Kit (Lonza, Basel, Switzerland) (Appendix A). iPSCs cells from passage 15 to 30 were frozen in 250 µL of cold Cryostem freezing medium (Clinisciences, Nanterre, France) and stored in liquid nitrogen.

### 2.3. Alkaline Phosphatase Assay

iPSC clones were tested at passage 5 for alkaline phosphatase (AP) activity, a marker of pluripotent stem cells. iPSC cultures were washed twice with DMEM/F12 and stained directly with 1× AP live stain solution (Molecular Probes, Thermo-Fisher Scientific) that contains a cell-permeable fluorescent substrate for alkaline phosphatase. After 30 min of incubation, cells were washed three times with DMEM/F12 for 5 min. To visualize the fluorescent-labeled colonies under a fluorescent microscope using a fluorescein isothiocyanate (FITC) filter, the observation was done no longer more than 90 min after staining.

### 2.4. Differentiation of Induced Pluripotent Stem Cells into Retinal Pigment Epithelial Cells

For differentiation into RPE cells, confluent iPSCs (defined as day 0) were cultured between passages 15 (p15) and p45, with a protocol adapted from Reichman et al. (2017) [25] and Slembrouck-Brec et al. (2018) [26]. Briefly, after 2 days of culture in the Essential 6™ Medium (Thermo-Fisher Scientific), the medium was replaced by a “proneural medium” composed of Essential 6™ Medium supplemented with 1% N2 supplement (Thermo-Fisher Scientific). The medium was changed every 2–3 days. At day 28, the medium was replaced by a “RPE medium” composed of the DMEM/F12 medium (Thermo-Fisher Scientific) supplemented with 1% N2 supplement, 1% MEM-NEAA and 1% penicillin-streptomycin and the medium was changed every 2–3 days. From day 42, identified pigmented patches, corresponding to iPSC-derived RPE (iRPE) cells, were excised with a needle and transferred onto vitronectin-coated 24-well plates, named passage 0 (P0). iRPE cells were cultured until confluence and passaged after TrypLE Express (Thermo-Fisher Scientific) cell-dissociation onto vitronectin-coated T25 (25 cm^2^) for further expansion, banking and analysis. iRPE cells were frozen at passage 1 or 2 in 250 µL of cold CryoStem freezing medium (Clinisciences) and stored in liquid nitrogen.

### 2.5. RT-PCR and Real-Time PCR Analyses

Total RNA was extracted using the RNeasy Plus Mini or Micro kit (Qiagen, Hilden, Germany) according to the manufacturer’s protocol and was quantified using a NanoDrop apparatus (Thermo-Fisher Scientific). For first-strand cDNA synthesis, 500 ng of total RNA were mixed with 500 ng random primers (Promega, Madison, WI, USA) and 10 mM dinucleotide triphosphates (dNTPs) (Invitrogen, Thermo-Fisher Scientific) and inactivated at 65 °C for 5 min. The reactions were supplemented with 5×-first strand buffer, 0.1 M DTT and the ribonuclease inhibitor (RNasin, Promega) and incubated for 2 min at room temperature before the addition of Superscript II Reverse Transcriptase (Invitrogen). Samples were incubated at room temperature for 10 min, then at 42 °C for 50 min before inactivation of the enzymatic reaction at 70 °C for 15 min.

For PCR analyses, cDNA was supplemented with 5× Green Go Taq Flexi Buffer (Promega), 25 mM MgCl_2_, dNTPs, 10 µM forward and reverse primers and GoTaq Flexi DNA Polymerase (Promega). PCR was performed using a thermocycler (Applied Biosystems 2720 Thermal Cycler, Thermo-Fisher Scientific) with 2 min denaturation at 95 °C, followed by 35 cycles of 30 sec denaturation at 95 °C, 30 sec annealing at 60 °C and 1 min extension at 72 °C, with 5 min final extension at 72 °C.

For real-time PCR analyses, cDNA (5 ng reverse-transcribed total RNA) was mixed with 10 µM forward and reverse primers and 1X Power SYBR Green (Invitrogen). Amplification was performed on a 7500 Fast real-time PCR System (Applied Biosystems) with a first cycle of 2 min at 50 °C and 10 min at 95 °C, followed by 40 cycles of 15 s at 95 °C and 1 min at 60 °C, ending by a dissociation stage. The expression of each gene was normalized by the expression of 18S ribosomal RNA housekeeping gene and quantified using the ΔCT method. The sequences of forward and reverse primers are listed in Appendix A.

### 2.6. Single Nucleotide Polymorphism Analysis

Genomic DNA was extracted from the iPSC clones and the corresponding fibroblasts, using DNeasy Blood and Tissue kit (Qiagen) according to the manufacturer’s instructions. Samples were analyzed by single nucleotide polymorphism (SNP) array using Illumina Infinium^®^ HumanCore-24v1-1 technique (Illumina, San Diego, CA, USA) by Integragen (Evry, France). The percentage of SNP concordance between each iPSC clone and the corresponding fibroblasts was determined by dividing the number of concordant genotypes by the number of total genotypes and multiplied by 100. Copy number variation (CNV) was plotted for each specimen as Log2 ratio plots (Illumina) (Appendix A).

### 2.7. Sanger Sequencing and Genotyping

Genomic DNA samples from each patient were genotyped for the SNP of interest by PCR amplification and Sanger sequencing by CD Genomics (Shirley, New York, NY, USA).

### 2.8. Immunofluorescence and Microscopy

iPSCs or iRPE cells were plated onto vitronectin-coated coverslips in 24-well plates. Confluent cells (3–5 days for iPSCs, and a minimum of 42 days for iRPE cells) were fixed in 4% paraformaldehyde for 15 min, washed in PBS and incubated at room temperature for 1 h with a permeabilizing blocking buffer composed of 0.5% bovine serum albumin (BSA) and 0.05% saponin in PBS. Primary antibodies (Appendix A) were incubated over night at 4 °C in permeabilizing blocking buffer, and washed three times with PBS, before incubation with the appropriate secondary antibody conjugated either with AlexaFluor 488 or 594 (Life Technologies, Thermo-Fisher Scientific) and with Hoechst 33342 (Invitrogen) in the blocking buffer for 1 h at room temperature. Coverslips were washed with PBS and mounted using Fluoromount (Invitrogen), and imaged with an inverted confocal microscope (Olympus, Tokyo, Japan).

### 2.9. Scorecard Assay of In Vitro iPSC Differentiation

The in vitro differentiation potential of iPSCs was assessed by their spontaneous differentiation into embryoid bodies (EBs). iPSCs were cultured in ultra-low adherent 6-well plates (Corning) for EB formation in Essential 6™ Medium. The medium was changed every 2–3 days, until day 20, when total RNA was extracted from EBs. Scorecard assay was performed by Thermo-Fisher Scientific on day 20-EBs compared to the corresponding iPSC clones. Scorecard values correspond to algorithm scores for the samples, showing upregulation or downregulation of the pluripotent (self-renewal), ectoderm, mesoderm and endoderm markers relative to a reference set of nine undifferentiated pluripotent stem cell lines.

### 2.10. Photoreceptor Outer Segments Phagocytosis

Photoreceptor outer segments (POSs) were purified from porcine eyes and covalently labeled with FITC as already described [27]. iRPE cells were plated onto vitronectin-coated black transparent 96-well plates (Corning) at a density of 80,000 cells/well and cultured for 62 days. Cells were incubated with FITC-labeled photoreceptor outer segments (FITC-POS) for 5 h at 37 °C and washed three times with PBS containing 0.2 mM CaCl_2_ and 1 mM MgCl_2_ (hereafter named PBS-CM, Thermo-Fisher Scientific). For detection of internalized photoreceptor outer segments, fluorescence of surface-bound FITC-POS was selectively quenched by incubating with 0.2% trypan blue in PBS-CM for 10 min. Cells were then fixed with ice-cold methanol for 10 min followed by rehydration in PBS-CM for 10 min and nuclei were stained with Hoechst 33342 1:1000 for 15 min at room temperature. Wells were imaged with Z-stack using an inverted confocal microscope and orthogonal views were obtained with ImageJ software (imagej.nih.gov/ij).

### 2.11. Western Blotting

Confluent cells (3–5 days for iPSCs, and a minimum of 42 days for iRPE cells) were washed with PBS and harvested, centrifuged 5 min at 800 round per min (rpm). The cell pellet was washed with PBS, and resuspended in radio immunoprecipitation assay (RIPA) buffer (Thermo Fisher-Scientific) supplemented with protease inhibitors (Sigma-Aldrich, St. Louis, MO, USA). After 30 min on ice and centrifugation for 20 min at 14,000 rpm at 4 °C, the supernatant was stored over night at −20 °C or at −80 °C for longer storage prior to analysis. Total proteins were quantified using the bicinchoninic acid assay (BCA) assay (Thermo-Fisher Scientific). Human post-mortem RPE/choroid cells were kindly provided by Valérie Fradot (Institut de la Vision) by harvesting cells in the eye-cup after removal of the retina with forceps and microdissection scissors. The cell pellet was resuspended in RIPA buffer. Constant protein amounts (10 µg) were loaded onto 9% SDS-PAGE gels, transferred onto polyvinylidene fluoride (PVDF) membranes (Millipore, Burlington, MA, USA) and saturated with 5% non-fat dry milk. Blots were probed with primary antibodies (Appendix A) and visualized with relevant horseradish peroxidase (HRP)-conjugated secondary antibodies and the ECL detection kit (Pierce, Thermo-Fisher Scientific).

### 2.12. Measurement of Transepithelial Lactate Transport

iRPE cells were seeded onto vitronectin-coated Transwell^®^ filters (Corning) in 12-well plates at a density of 125,000 cells/well. Transepithelial electrical resistance (TEER) was monitored once or twice a week using an epithelial volt/ohmmeter (EVOM™ World Precision Instrument, Friedberg, Germany). Net TEER values (Ω.cm^2^) were calculated by subtracting the value of a Transwell insert without cells from the mean of eleven wells from the same plate, and by multiplying the result by the surface area of the Transwell filter. Cells were cultured until reaching a plateau (between day 90 and 110), indicating that the iRPE cells were mature and polarized and that iRPE barrier was tight. L-lactate transport assay was performed using RPE Ringer’s solution, composed of 116.5 mM NaCl, 26.2 mM NaHCO_3_, 5 mM KCl, 0.5 mM MgCl_2_, 1.8 mM CaCl_2_, 1 mM l-carnitine and HEPES-*N*-methyl-d-glucamine (12 mM HEPES dissolved in deionized H_2_O and titrated to pH 7.4 with *N*-methyl-d-glucamine), as described previously [28]. Glucose or other substrates were excluded from the solution to avoid any lactate production by the tested cells. Ringer’s solution was equilibrated with 5% CO_2_ before sterile filtration. iRPE cells on Transwell filters were transferred to a new 12-well plate and washed with Ringer’s solution (the apical and basal chambers). The apical chamber was then replaced with 0.5 mL Ringer’s solution and incubated at 37 °C and 5% CO_2_ for 15 min. Another 12-well plate was preincubated in parallel with 1.5 mL per well Ringer’s solution at 37 °C and 5% CO_2_ for 15 min. Ringer’s solution from the basal chamber was removed, and the apical chambers of each Transwell filter were replaced with 1.5 mL Ringer’s solution containing 5 mM l-lactate (Sigma). Before transferring to the preincubated plate, 100 µL were collected from basal chambers of three independent wells, corresponding to the t0 time point. All Transwell chambers were then transferred to the preincubated plate, and 100 µL were collected from all basal chambers at 10, 30, 45 and 60 min for the evaluation of lactate concentration using an enzymatic colorimetric lactate assay kit (Sigma-Aldrich, MAK064) following the manufacturer’s instructions.

### 2.13. Statistical Analysis

Quantitative data are the means ± standard deviation (SD) of at least three independent experiments. RT-PCR gel images and Western blots are representative of at least two independent experiments. GraphPad Prism 6 (GraphPad software Inc., La Jolla, CA, USA) was used for statistical analyses. For the L-lactate transport assay, Dunnett’s multiple comparisons test was used. The following symbols were used: * *p* ≤ 0.05 and *** *p* ≤ 0.001.

## 3. Results

### 3.1. Generation and Characterization of Patient-Derived iPSCs

Since *SLC16A8* is specifically expressed by the RPE, the function of the lactate transporter MCT3 cannot be studied in other tissues. Tissue-specific pattern analysis of mRNA expression in the mouse indicated that *Slc16a8*, encoding the lactate transporter MCT3, was restricted to the RPE (Appendix A). Induced pluripotent stem cells were generated from skin biopsies of patients from the AMD cohort from the Centre d’Investigation Clinique (CIC) of the Centre Hospitalier National Ophtalmologique des Quinze-Vingts (CHNO-XV-XX) in Paris carrying or not the risk allele at the locus rs77968014 in the *SLC16A8* gene. Patient #3130 was a non-carrier control for rs77968014 without the two major alleles reported to predispose to AMD, namely *CFH* and *ARMS2/HTRA1* genes [11]. Patient #4024 was the only patient from the cohort of 1,154 patients who was homozygous for the risk allele at rs77968014 in the *SLC16A8* gene. Spectral domain optical coherence tomography (SD-OCT) was used to diagnose adult-onset foveomacular vitelliform dystrophy (AFVD) patients. It facilitated confirmation of the subretinal location of vitelliform lesions, the morphometric studies of the lesions, and accompanying findings in the retina and RPE, and allowed a clear discrimination between AFVD and AMD (Figure 1) [29]. This patient was also carrying two copies of the risk allele rs10490924-T at the *ARMS2/HTRA1* locus. Since the serine peptidase HTRA1 is mainly expressed by mononuclear phagocytes in the eye, it probably does not interfere here with the *SLC16A8* alleles expressed specifically by the RPE [10].

Skin fibroblasts were reprogrammed into iPSCs using the non-integrative Sendai virus and reprogramming factors *OCT3/4*, *SOX2*, *KLF4* and *cMYC*. iPSC colonies started emerging around day 10, and 8–15 clones per patient were selected and picked up. Some clones were rapidly lost due to incomplete reprogramming, but all the remaining ones were displaying alkaline phosphatase activity at passage 5 and were further characterized (Figure 2A). Between passages 13 and 14, RT-PCR analysis of iPSC clones confirmed that iPSC clones c2, c6, c7 and c8 of control patient #3130 had lost ectopic expression of reprogramming factors KLF4, OCT3/4, SOX2, (KOS) and cMYC, whereas cMYC and the Sendai virus (SeV) genome were still detected in clone c1, which was consequently eliminated (Figure 2B). For patient #4024, iPSC clones c1, c5 and c7 show an absence of genomic integration of reprogramming factors after 16–18 passages, whereas clones c3 and c6 were eliminated (Figure 2C). Copy number variation (CNV) plot analysis using single nucleotide polymorphism (SNP) array confirmed the genomic integrity of the iPSC clones selected above, namely clones #3130-c6, c7, c8 and #4024-c1, c5 and c7 (Appendix A). The parenthood of iPSC clones to their respective fibroblasts was superior to 99.91% (Figure 2D). The expression of endogenous stem cell markers lin-28 homolog A (*LIN28A*) and POU class 5 homeobox 1 (*POU5F1*) was analyzed by quantitative PCR analysis. These markers could not be detected in fibroblasts but were present in all selected iPSC clones from both patients (Figure 2E). The presence of two copies of the risk allele at rs77968014 (c.214+1GC) in the *SLC16A8* gene in #4024 patient was confirmed by Sanger sequencing (Figure 2F). Sequencing also confirms that control iPSCs derived from patient #3130 were non-carrier for the rs77968014-G risk allele. Using immunocytochemistry, we observed that all iPSC clones displayed specific ESC-like morphology, with large nuclei and tightly packed colonies expressing pluripotency markers such as transcription factors OCT4 and NANOG in the nucleus and proteins SSEA4 and TRA1-81 at the cell surface (Figure 2G). The capacity of these iPSC clones to self-differentiate into the three germ layers, ectoderm, mesoderm and endoderm, and to lose their iPSC-specific self-renewal capacity, was demonstrated by embryoid body (EB) formation combined with score-card analysis (Figure 2H and Appendix A). Any given score represents a statistical comparison of the gene expression profile of each specimen in the three germ layers, to that of the undifferentiated reference set. iPSC clones #3130-c6, c7 and c8, scoring positive for self-renewal markers and negative for germ layer markers, were selected for further studies, while clone c8 was eliminated. For the splicing iPSCs, clones #4024-c1 and c7 were validated, but not clone c5, which displays a borderline gene expression score for ectoderm germ layer in iPSCs. All iPSC cultures were mycoplasma-free prior to differentiation (Appendix A).

### 3.2. Differentiation of iPSCs into iRPE Cells

We induced the differentiation of selected iPSCs towards RPE by maintaining iPSCs for two days in culture medium containing fibroblast growth factor 2 (FGF2), before switching the culture to a proneural medium containing N2-supplement that promotes neuroectoderm induction. After 28 days, the medium was replaced by culture medium that favors RPE differentiation (see Materials and Methods). We monitored the appearance of pigmented patches, a visible marker of RPE differentiation. Pigmented iPSC-derived RPE (iRPE) patches were manually picked and separately amplified. Some iPSC clones did not have the capacity to differentiate into enough RPE patches and were consequently discarded. iPSC clones #3130-c7 and #4024-c1 and -c7 gave rise to numerous RPE patches. We found using real-time PCR that iRPE cells from these clones have lost the expression of pluripotency markers *LIN28A* and *POU5F1* (Figure 3A). Conversely, all iRPE cells express RPE RNA markers such as the retinal pigment epithelium–specific 65-kDa protein (*RPE65*), bestrophin 1 (*BEST1*) and the microphthalmia-associated transcription factor (*MITF*), which are not expressed by iPSCs (Figure 3B). The maturation of iRPE does not seem to be complete when compared to adult human RPE. It has been reported that iRPE gene expression profiles are more similar to that of human fetal than adult RPE [30]. Moreover, beyond visible pigmentation (Figure 3C, upper panel), immunocytochemical analysis revealed the expression of MITF in the nucleus and zonula occludens-1 (ZO-1) at tight junctions, evidencing the polygonal morphology of the cells. Orthogonal views show that ezrin was located at the apical side of iRPE cells from both clones #3130-c7 and #4024-c7 at day 56 (Figure 3C, lower panels).

To assess whether iRPE cells were functional, their capacity to phagocytose FITC-labeled photoreceptor outer segments (POSs) was examined within a period of 5 h. FITC-conjugated POS were added to the cells and internalized POS were detected after surface-bound FITC-conjugated POS quenching by trypan blue five hours after. As seen in confocal images with an orthogonal view, iRPE cells with two copies of the splicing allele in the *SLC16A8* phagocytize FITC-conjugated POS, even if internalized POS are less numerous than for the control iRPE (Figure 3D). Altogether, these results demonstrated that these differentiated cell lines exhibited a functional RPE phenotype.

### 3.3. A Rare Splice Variant in the SLC16A8 Gene Leads to a Lactate Transport Deficit

AMD risk alleles within the *SLC16A8* gene could alter transepithelial transport of lactate through the RPE since it involves two distinct transporters MCT1 (*SLC16A1*) on the apical side, toward photoreceptors, and MCT3 (*SLC16A8*) on the basal side, toward the choroid [16]. A deficit in MCT3 could trigger an increase of lactate concentration in the outer retina counteracting aerobic glycolysis of photoreceptors. This mechanism likely explains the impaired photoreceptor function and cone-damage observed in the *Slc16a8^-/-^* mouse [12]. The absence of the GT consensus sequence of human 5′ donor sites in the *SLC16A8* gene is predicted to affect the splicing process (Figure 4A) [31]. In order to analyze the effect of the rs77968014-G allele on splicing, we examined the splicing of *SLC16A8* by iRPE using RT-PCR by designing specific primers (Figure 4B). As expected, control iRPE cells #3130 and post-mortem human RPE cells express a mRNA resulting from the correct splicing of the gene, whereas the mRNA from iRPE cells of patient #4024 was larger, due to the presence of intron 2 in the amplification product resulting from the absence of splicing of intron 2, revealing intron retention (Figure 4C).

We used Western blotting to analyze the consequence of intron retention on the expression of MCT3 by iRPE cells from patient #4024. MCT3 is expressed by control iRPE cells and post-mortem human RPE cells, but not by the corresponding iPSCs (Figure 4D). By contrast, even if iRPE cells from patient #4024 with a splicing defect express RPE markers at the same level as control iRPE cells, MCT3 expression was not detected. However, the absence of MCT3 expression does not impact the differentiation status of iRPE cells as two markers of RPE maturation RPE65 and BEST1 are expressed irrespective of the allele rs77968014-G. The reduced intensity of the larger RT-PCR product of #4024 iRPE cells mRNA (Figure 4C) suggests a mechanism of nonsense-mediated mRNA decay aiming at eliminating the mRNA transcript containing a premature stop codon [32]. Lactate transporters expression and localization were also analyzed by immunocytochemistry and confocal orthogonal projections. The lactate transporter MCT1, encoded by the *SLC16A1* gene, is localized at the apical side of both control #3130 and splicing #4024 iRPE cells, above ZO-1, a marker of tight junctions (Figure 4E). Conversely MCT3 is localized on the basal side of the RPE in control iRPE cells from patient #3130, below the tight junctions as previously reported in the mouse [12]. The expression of MCT3 is not confined to the basolateral surface as observed in vivo [12], but immunocytochemical analysis resembles what was previously observed in vitro [33]. In accordance to Western blotting analysis, MCT3 was not detected by immunocytochemistry in iRPE cells carrying the *SLC16A8* splicing allele.

The transepithelial electrical resistance of the iRPE cultures was recorded every week until reaching a plateau and shows that the splicing #4024 iRPE cells are slightly more compact (Figure 4F). Furthermore, to test whether iRPE cells with a splicing defect were able to transport lactate, a lactate assay was performed in Transwell chambers (Figure 4G). Five millimeter L-lactate was added on top of mature and polarized iRPE monolayers seeded 90–110 days before in the upper chamber of Transwell inserts. Lactate that has been transported from the upper chamber by the apical lactate transporter MCT1, through the RPE and then into the medium of the lower chamber by MCT3 at the basal side was quantified using a colorimetric assay. Its concentration increased over 60 min. We observed that the lactate transport rate was significantly lower for the splicing iRPE cells that did not express MCT3, evidencing a lactate transport deficit in these cells (Figure 4H).

We show that an iRPE cell line with two copies of the splicing allele rs77968014-G results in the absence of MCT3 expression, both by Western blotting and immunocytochemistry. A bioinformatic analysis predicts the production of four hypothetic mRNAs based on the potential use of cryptic donor sites. Without the use of any cryptic donor site, the translated protein would be 472 amino acids (aa) long with the insertion of the peptide corresponding to the translation of intron 2 connected to in-frame exon 3 (Figure 5A). The use of alternative cryptic sites would result in a protein of 325 (cryptic site 7), 214 (cryptic site 11) or 265 (cryptic site 17) aa, including the sequence of exon 3, and would be out of frame as compared to the original MCT3 protein. The identity of these cryptic donor sites proteins would be limited to the N-terminal 72 residues of MCT3. The RT-PCR product of the *SCL16A8* mRNA without the use of any cryptic site would be 666 bp (1164–498) and fits with our observation (Figure 4C). The mutant MCT3 protein is the result of intron retention and the addition of 209 aa in the middle of the second transmembrane domain. The antibody we used was raised against a C-terminal peptide of human MCT3 residues GEPTEPEIEARPRLAAAESV [34], so we could have detected the putative mutant protein of 713 residues, but we did not (Figure 4D,E). The introduction of 209 aa in transmembrane 2 of MCT3, a 504 aa protein with 12 transmembrane domains, modifies profoundly the predicted structure of the mutated protein (Figure 5B). The existence of a 209 aa domain as an intracellular loop would destabilize the mutant protein that would be degraded by the cell, even if the 209 aa encompass an additional transmembrane domain (Intron TM) [35]. Our analysis of the differentiation of the #4024 iPSCs into an iRPE cell line suggests that the putative mutated protein does not harbor a gain of function damageable to the cell as it maturates correctly and does not show evidence of an abnormal morphology, apart from the smaller cell volume of iRPE cells (Figure 4E). Our conclusion is that MCT3 with the rs77968014-G allele had a large deficit in lactate transport activity but it was more equivalent to an hypomorphic allele with a potential residual activity of the mutated allele expressed at a very low level than to a real knock-out allele.

## 4. Discussion

The transepithelial lactate transport of the iRPE cell line with two copies of the splicing allele rs77968014-G was statistically reduced as compared to the non-carrier iRPE cell line, but an increase in lactate concentration was observed within the one-hour assay (Figure 4H). One could speculate that MCT1 could compensate for MCT3 in this artificial setting. The immunocytochemical analysis show that MCT1 was localized to the apical side of the monolayer of the #4024 iRPE culture, suggesting that MCT1 did not compensate for MCT3 (Figure 4E). We could not exclude either that another lactate transporter (to be investigated) could be expressed at the basal membrane of RPE cells to compensate for MCT3. The absence of inhibitors specific to MCT3 did not allow us to address this question without ambiguity [36]. Alternatively, the effect of the MCT3 mutation on lactate transport might be more pronounced at other lactate concentrations. One possible alternative explanation is that the culture contains patches of loose junctions even if the transepithelial electrical resistance was monitored and that the tight junction marker ZO-1 aspect is rather uniformly distributed in the culture (Figure 4E). Another alternative is that part of the lactate measured in the lower chamber of the Transwell originates from iRPE cell metabolism. This would make sense if we consider that in vitro, iRPE cells are maintained in glucose medium, even if glucose or other substrates were excluded from the iRPE solution before the assay was performed (see Materials and Methods). The use of fluxomics could potentially improve the measure of transepithelial lactate transport [37].

Nevertheless, the rare variant rs77968014-G (9.1 × 10^−6^) could not be genetically associated with AMD since it did not reach the genome-wide significance of 5 × 10^−8^ [13]. The allele rs8135665-T that is genetically associated with AMD [11] and rs77968014-G are likely two independent signals (Lars Fritsche personal communication). The unique donor (#4024) homozygous for this causative allele is affected by adult-onset foveomacular vitelliform dystrophy (AFVD) (Figure 1). AFVD is characterized by a solitary, oval, slightly elevated yellowish subretinal lesion of the fovea that is similar in appearance to the vitelliform or egg-yolk stage of the Best disease, but distinct [29,38,39]. Patients usually become symptomatic in the fifth or sixth decade of life with a decrease of their visual acuity. Few patients with age-related macular degeneration (AMD) have retinal lesions that resemble AFVD, which questions whether AFVD is a dystrophy or a subtype of AMD [40]. We think that patient #4024 was recruited to the AMD cohort because of confusion in the retinal phenotype [40]. Serendipitously, it is quite possible that the loss of function of the lactate transporter MCT3 is the cause of AFVD for this patient.

## 5. Conclusions

The loss of transport function of the hypomorphic allele in the *SLC16A8* gene provides a methodologic frame to the study of other *SLC16A8* alleles genetically associated with AMD. This study paves the way to the investigation of transepithelial lactate transport in AMD. 

## Figures and Tables

**Figure 1 cells-10-00179-f001:**
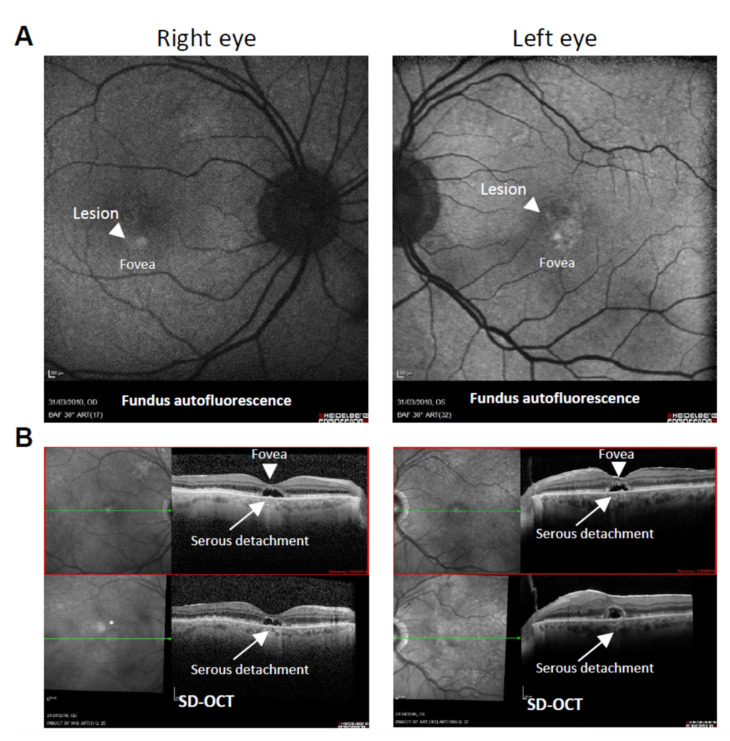
Clinical characteristics of the adult onset foveal vitelliform dystrophy (AFVD) patient #4024 carrying two copies of the *SLC16A8* allele rs77968014-G. (**A**) Fundus autofluorescence imaging shows a bilateral lesion of the fovea. (**B**) Spectral domain optical coherence tomography (SD-OCT) shows a bilateral dome-shaped hyperreflective subretinal lesion (serous lesion) corresponding to the lesion seen by autofluorescence in (**A**). Scale bar: 200 µm.

**Figure 2 cells-10-00179-f002:**
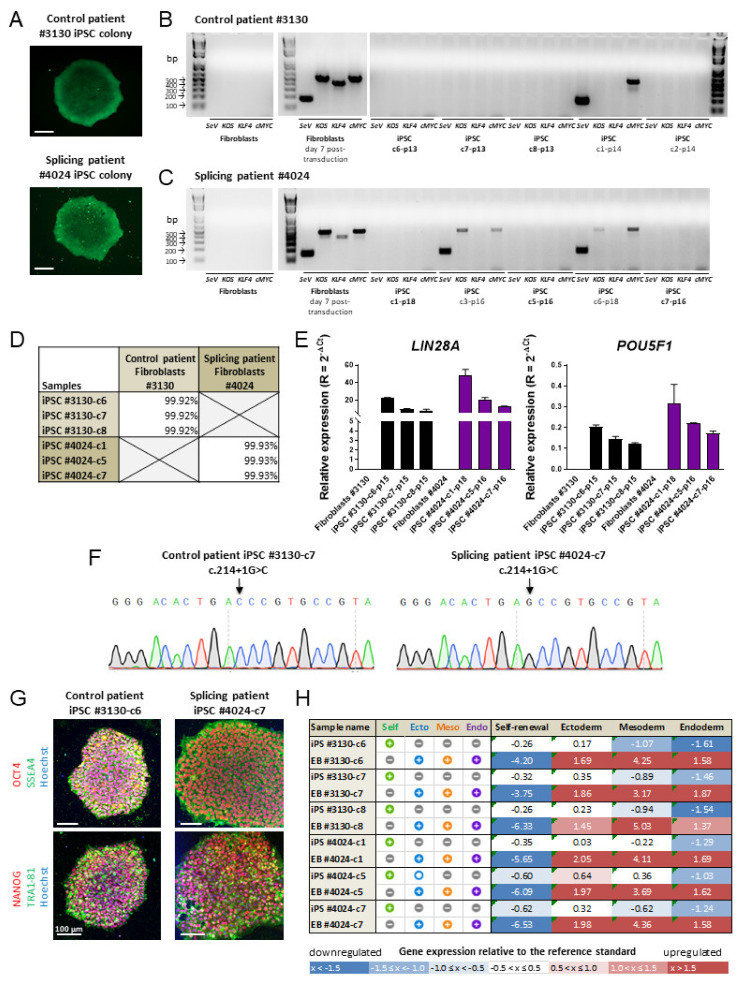
Reprogramming and characterization of patient-derived induced pluripotent stem cells. (**A**) Alkaline phosphatase activity of induced pluripotent stem cell (iPSC) colonies at passage 5. Scale bar: 100 µm; (**B**,**C**) RT-PCR analyses of (**B**) #3130 control and (**C**) #4024 splicing patient’s fibroblasts before and 7 days post-transduction and selected iPSC clones after several passages (p13 to p18); (**D**) percentage of single nucleotide polymorphism (SNP) concordance between each iPSC clone and corresponding fibroblasts; (**E**) quantitative PCR analysis of specific stem cell markers *LIN28A* and *POU5F1*. Error bars represent technical replicates of samples from a single passage; (**F**) Sanger sequencing confirming the presence of the risk allele in the splicing patient iPSCs; (**G**) immunocytochemical analysis of iPSCs for pluripotent nuclear proteins OCT4 and NANOG, and surface markers SSEA4 and TRA1-81. Scale bar: 100 µm; (**H**) Scorecard analysis with values comparing the expression profile of self-renewal, ectoderm, mesoderm and endoderm genes in iPSCs and in vitro-formed embryoid bodies relative to a reference iPSC standard set.

**Figure 3 cells-10-00179-f003:**
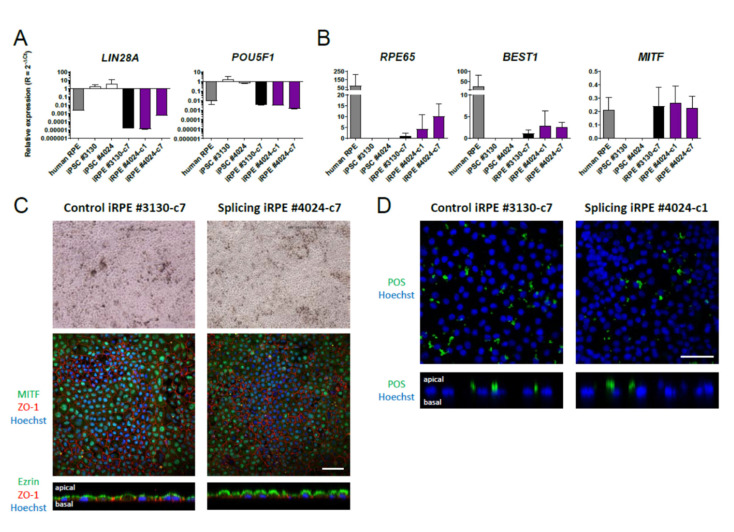
Characterization of iPSC-derived retinal pigment epithelium. (**A**) Quantitative RT-PCR analysis of the expression of stem cell markers *LIN28A* and *POU5F1*. The data were normalized to 18S rRNA expression; (**B**) quantitative RT-PCR analysis of the expression of retinal pigment epithelium (RPE) specific markers *RPE65*, *BEST1* and *MITF.* The data were normalized to 18S rRNA expression; (**C**) phase-contrast images of pigmented iPSC-derived RPE (iRPE) cells (upper panel) and immunofluorescent staining (lower panels) of RPE markers MITF in the nucleus, ZO-1 at tight junctions with orthogonal view of ezrin and (**D**) confocal images with orthogonal view of internalized FITC-conjugated POS in iRPE cells from clones #3130-c7 and #4024-c1 after 5 h of phagocytosis. Nuclei were stained with Hoechst 33342. Scale bar: 50 µm.

**Figure 4 cells-10-00179-f004:**
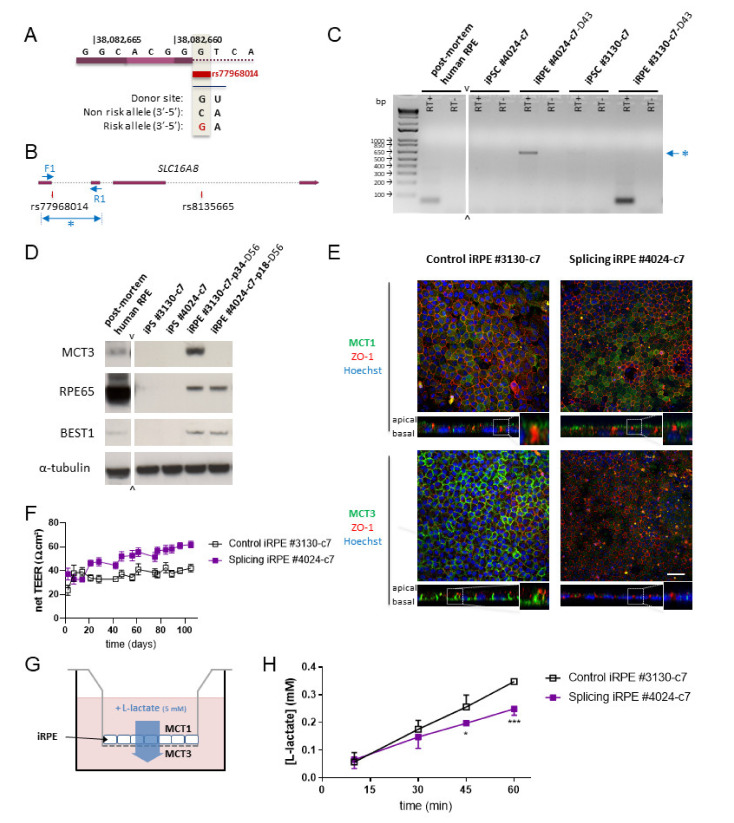
Lactate transport deficit in iRPE cells from patient homozygous for the splicing risk allele in *SLC16A8* gene. (**A)** Presence of a rare 5′ putative splice-site variant in age-related macular degeneration (AMD) at rs77968014 in the *SLC16A8* gene; (**B**) position of the primers used to study the splicing of the gene in iRPE cell lines; (**C**) RT-PCR analysis of *SLC16A8* gene splicing, revealing intron retention for the iRPE cell line #4024 (at day 43 of differentiation) and normal splicing of the gene in control iRPE cell line #3130 (at day 43) and post-mortem human RPE cells, *SLC16A8* being absent in iPSCs. RT+ and RT-, with and without reverse transcriptase, respectively and (**D**) whole cell lysates of iPSCs, iRPE (at day 56) and post-mortem human RPE were analyzed by Western blotting for the expression of MCT3, RPE65 and BEST1. α-tubulin was used as a loading control; (**E**) immunocytochemical analysis with orthogonal projections of MCT1 and ZO-1 (upper panel), and MCT3 and ZO-1 (lower panel) in control and splicing iRPE cells at day 56. Scale bar: 50 µm; (**F**) transepithelial electrical resistance monitoring of iRPE monolayers onto Transwell chambers up to 105 days; (**G**) lactate transport assay experimental setting and (**H**) measurement of lactate concentration transported within 60 min across control and splicing iRPE monolayers at day 105.

**Figure 5 cells-10-00179-f005:**
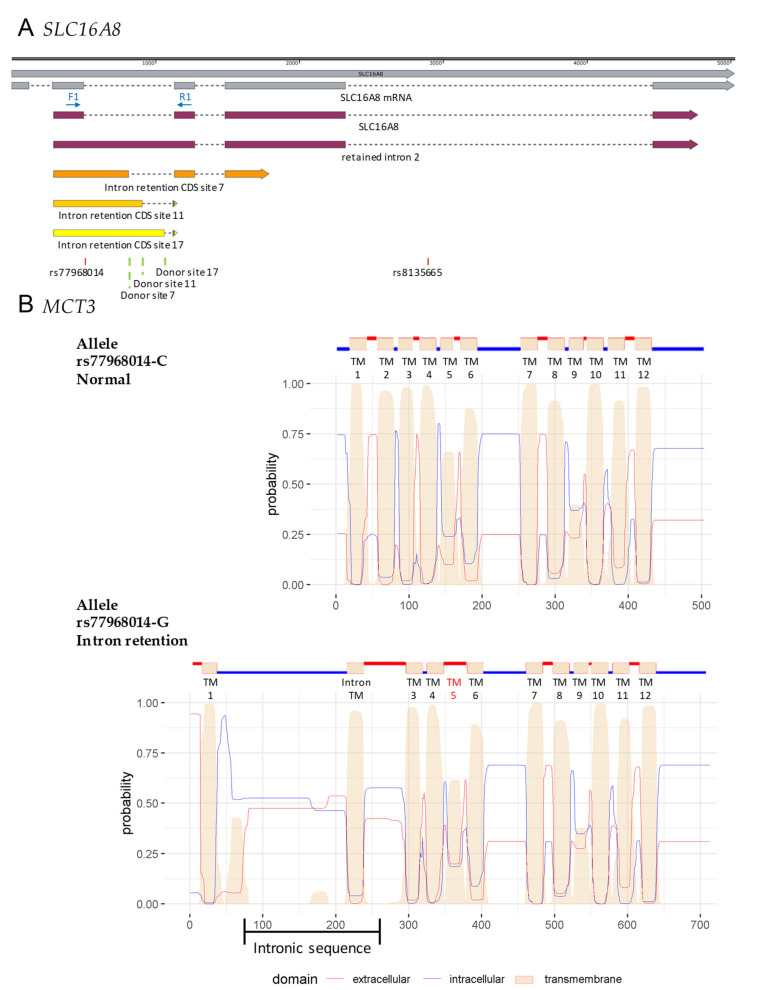
Bioinformatic analysis of potential cryptic donor sites for the allele rs77968014-G. (**A**) We introduced the rs77968014 SNP in the sequence of the human *SLC16A8* gene and analyzed the potential donor sites that could be revealed with the Human Splicing Finder 3.1. We found 3 sites above the threshold (donor site 3, 11 and 17) among the 20 potential donor sites. We used SnapGene^®^ software (from GSL Biotech; available at snapgene.com) to visualize the 3 sites and the resulting coding sequences (CDS). (**B**) Prediction of transmembrane helices in proteins with TMHMM (http://www.cbs.dtu.dk/services/TMHMM/) to study the predicted protein.

## Data Availability

No new data were created or analyzed in this study. Data sharing is not applicable to this article.

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
