# Peer review of "A Splice Variant in *SLC16A8* Gene Leads to Lactate Transport Deficit in Human iPS Cell-Derived Retinal Pigment Epithelial Cells"

_cells, 2021, doi:10.3390/cells10010179_

Round 1
Reviewer 1 Report
The study is interesting and well-presented. The methods are presented in detail and this is highly desirable considering the amount of results presented and the design of the study. My only concerns deal with the conclusive parts of the discussion and conclusions, which appear a little bit vague in my opinion. Perhaps, avoiding open questions within the text would be helpful to make the conclusions less ambiguous.
Author Response
The study is interesting and well-presented. The methods are presented in detail and this is highly desirable considering the amount of results presented and the design of the study. My only concerns deal with the conclusive parts of the discussion and conclusions, which appear a little bit vague in my opinion.
We have modified the conclusion as followed (line 500):
The loss of transport function of the hypomorphic allele in the SLC16A8 gene provides a methodologic frame to the study of other SLC16A8 alleles genetically associated with AMD. This study paves the way to the investigation of transepithelial lactate transport in AMD.
Perhaps, avoiding open questions within the text would be helpful to make the conclusions less ambiguous.
We have incorporated the first chapter of the discussion in the result section as asked by reviewer We have removed the three introducing questions from the text.
Reviewer 2 Report
Klipfel et al. present very interesting findings producing iPSCs from patients with or without splice variant SLC16A8 gene. Thorough characterization was performed which validate the successful differentiation of the iPSCs into RPE cells. Using these induced iRPE cells, the authors show the impact of the splicing variant on the expression of MCT3 as well as on lactate transport. I have no hesitation in recommending this work for publication. However, prior to that the following should be taken into consideration and addressed:
Figure 2E: Please include error bars.
Fig. 3B: RPE65 and BEST1 expression in iRPE #3130-c7 do indicate the differentiation of iPSC #3130-c7 into RPE, but the expression is drastically lower compared to that in hRPE. This is also apparent at protein level in figure 4D. Please comment.
Fig. 4C: hRPE and iRPE #3130-c7-D43 RT-PCR bands are not at the same size. Alignment issue?
Fig. 4D: Why does MCT3 blot result in multiple bands from the pm hRPE samples?
Fig. 4G: Did you also try to perform knockdown of MCT3 in control iRPE #3130-c7 cells and then measure the lactate transport?
Figure 5: as well as its related text should rather be moved to the Results section.
Line 61 : Expand BSG1 (Basigin-1)
Line 266: ..cannot be studies in other tissues
Line 275: Please expand AFVD already here
Line 277: Cite figure 1.
Line 295: LIN28A and POU5F1 that are expanded in line 338 should be expanded here.
Line 300: ESC-like morphology (for consistency)
Line 305: Figure S2A-B should be figure S3A-B
Line 307-309: iPSC clones #3130-c6, c7 and c8, scoring positive for self-renewal markers and negative for germ layer markers, were selected for further studies, while clone c8 was eliminated..
Line 343: Please expand ZO-1 (Zonula occludens-1)
Line 359: ..splicing allele in the SLC16A8 phagotize FITC-conjugated POS.. phagocytize
Fig. 4D: The uncropped blots may be removed from the figure.
Line 416: .. splicing iRPE cells that no not express MCT3..
Author Response
Klipfel et al. present very interesting findings producing iPSCs from patients with or without splice variant SLC16A8 gene. Thorough characterization was performed which validate the successful differentiation of the iPSCs into RPE cells. Using these induced iRPE cells, the authors show the impact of the splicing variant on the expression of MCT3 as well as on lactate transport. I have no hesitation in recommending this work for publication. However, prior to that the following should be taken into consideration and addressed:
Figure 2E: Please include error bars.
We have added the errors bars and specified in the figure legend (line 324): Error bars represent technical replicates of samples from a single passage.
Fig. 3B: RPE65 and BEST1 expression in iRPE #3130-c7 do indicate the differentiation of iPSC #3130-c7 into RPE, but the expression is drastically lower compared to that in hRPE. This is also apparent at protein level in figure 4D.
We have reconstructed figure 4, since part of the panel D corresponds to raw data asked by the journal and introduced by mistake in the figure for publication.
Please comment
We have added the following sentence with its citation in the manuscript (line 343): The maturation of iRPE does not seem to be complete when compared to adult human RPE. It has been reported that iRPE gene expression profiles are more similar to that of human fetal than adult RPE (Smith et al., Stem cell reports 12, 1342 (2019).
Fig. 4C: hRPE and iRPE #3130-c7-D43 RT-PCR bands are not at the same size. Alignment issue ?
We have re-aligned the two panels correctly.
Fig. 4D: Why does MCT3 blot result in multiple bands from the pm hRPE samples?
We have reconstructed figure 4D (see above). We do not know the nature of the MCT3 species with slow migration but this phenomenon is sometimes observed for proteins with multiple transmembrane domains as rhodopsin.
Fig. 4G: Did you also try to perform knockdown of MCT3 in control iRPE #3130-c7 cells and then measure the lactate transport?
No, since we did not know MCT3 turnover over that of siRNA.
Figure 5: as well as its related text should rather be moved to the Results section.
Done (see answer to Reviewer 1 above).
Line 61 : Expand BSG1 (Basigin-1)
Done
Line 266: ..cannot be studies in other tissues
Studied (line 268)
Line 275: Please expand AFVD already here
Done (line 277)
Line 277: Cite figure 1.
Done (line 279)
Line 295: LIN28A and POU5F1 that are expanded in line 338 should be expanded here.
Done (line 297, and removed line 340)
Line 300: ESC-like morphology (for consistency)
Done (line 303)
Line 305: Figure S2A-B should be figure S3A-B
Done (line 308)
Line 307-309: iPSC clones #3130-c6, c7 and c8, scoring positive for self-renewal markers and negative for germ layer markers, were selected for further studies, while clone c8 was eliminated.
Done (lines 310-312)
Line 343: Please expand ZO-1 (Zonula occludens-1)
Done (line 347)
Line 359: ..splicing allele in the SLC16A8 phagotize FITC-conjugated POS.. phagocytize
Done (line 363)
Fig. 4D: The uncropped blots may be removed from the figure.
We have reconstructed figure 4D (see above).
Line 416: .. splicing iRPE cells that no not express MCT3.
do not (line 422)
Reviewer 3 Report
The authors explore the metabolic consequences of a mutation in the gene encoding an RPE-specific lactate transporter, MCT3. The mutation was thought to increase risk for AMD (or as the authors point out a disease state more related to Adult Onset Foveal Vitelliform Dystrophy). They induced pluripotent stem cells from a patient homozygous for the mutation (along with cells from controls) and differentiated them into RPE-like cells. The mutation alters mRNA splicing and the MCT3 protein in the differentiated cells is undetectable by immunological methods. The mutation did not substantially alter the appearance or expression patterns of the cells, but it did slightly, but significantly decrease lactate transport by the cells. The authors reasonably discussed the significance of their finding in the context of RPE function and diseases.
The authors should address the following:
- FIG 4D. Please describe what is loaded in the lane 7 marked "other".
- 4E is puzzling. MCT3 labeling (green) appears to be at the junctions of the cells. In the orthogonal view the green signal does not appear to be confined to the basolateral surface (instead it seems to extend significantly towards the apical surface – probably along the cell-cell contacts?). Along the basal surface it does not appear to be uniformly distributed. Please discuss this unexpected (different than in RPE tissue from eyes shown in Fig. 1C of reference 12) distribution of MCT3.
- Fig. 4G. It seems surprising that there is any lactate transport at all via the monolayers without MCT3. Does this mean that there are other lactate transporters or does it mean that the monolayer is leaky. The authors discuss this later, but it would be good in the results section to report the transpeithelial resistance for these preparations. Also, if there are multiple transporters, or if MCT1 or some other transporter is affected the concentration dependence (Km) for lactate transport might be different. In other words, the effect of the MCT3 mutation on lactate transport might be more pronounced at other lactate concentrations. Also, is the residual lactate transport affected by MCT inhibitors (general inhibitors and isoform specific ones?)
- Please rewrite the sentence on lines 441-443. The start of the sentence seems to say that the allele is null for lactate trasnport activity, but then the second part of the sentence seems like it contradicts the idea that there would be no lactate transport by this mutant protein. Also, if there is some residual MCT3 protein expressed shouldn't it be detected in the westerns and ICC?
Author Response
The authors explore the metabolic consequences of a mutation in the gene encoding an RPE-specific lactate transporter, MCT3. The mutation was thought to increase risk for AMD (or as the authors point out a disease state more related to Adult Onset Foveal Vitelliform Dystrophy). They induced pluripotent stem cells from a patient homozygous for the mutation (along with cells from controls) and differentiated them into RPE-like cells. The mutation alters mRNA splicing and the MCT3 protein in the differentiated cells is undetectable by immunological methods. The mutation did not substantially alter the appearance or expression patterns of the cells, but it did slightly, but significantly decrease lactate transport by the cells. The authors reasonably discussed the significance of their finding in the context of RPE function and diseases.
The authors should address the following:
FIG 4D. Please describe what is loaded in the lane 7 marked "other".
We have reconstructed figure 4, since part of the panel D corresponds to raw data asked by the journal and introduced by mistake in the figure for publication.
4E is puzzling. MCT3 labeling (green) appears to be at the junctions of the cells. In the orthogonal view the green signal does not appear to be confined to the basolateral surface (instead it seems to extend significantly towards the apical surface – probably along the cell-cell contacts?). Along the basal surface it does not appear to be uniformly distributed. Please discuss this unexpected (different than in RPE tissue from eyes shown in Fig. 1C of reference 12) distribution of MCT3.
In the revised figure 4E, we have shifted the inset for MCT3 to show a little more the basolateral, labelling. We had indeed indicated in the text that MCT3 is localized “below the tight junctions”(line 409). We have added the following sentence and its citations (lines 409-411): The expression of MCT3 is not confined to the basolateral surface as observed in vivo (Daniele et al., American journal of physiology. Cell physiology 295, C451, 2008), but immunocytochemical analysis resembles what was previously observed in vitro (Castorino et al., Traffic 12, 483 (2011).
Fig. 4G. It seems surprising that there is any lactate transport at all via the monolayers without MCT3. Does this mean that there are other lactate transporters or does it mean that the monolayer is leaky. The authors discuss this later, but it would be good in the results section to report the transepithelial resistance for these preparations.
We have added in the figure 4F the transepithelial electrical resistance for these preparations, and added the following sentence in the legend (lines 392-393): (F) Transepithelial electrical resistance monitoring of iRPE monolayers onto Transwell chambers up to 105 days The materials and methods section was completed (lines 237-240): Net TEER values (Ω.cm²) were calculated by substracting the value of a Transwell insert without cells from the mean of eleven wells from the same plate, and by multiplying the result by the surface area of the Transwell filter. We have added in the text the following sentence (lines 414-415): The transepithelial electrical resistance of the iRPE cultures was recorded and shows that the splicing #4024 iRPE cells are slightly more compact (Figure 4F).
Also, if there are multiple transporters, or if MCT1 or some other transporter is affected the concentration dependence (Km) for lactate transport might be different. In other words, the effect of the MCT3 mutation on lactate transport might be more pronounced at other lactate concentrations.
We have added the following sentence on line 475 of the discussion: Alternatively, the effect of the MCT3 mutation on lactate transport might be more pronounced at other lactate concentrations.
Also, is the residual lactate transport affected by MCT inhibitors (general inhibitors and isoform specific ones?).
We have added the following sentence and its citation on line 474 of the discussion: The absence of inhibitors specific to MCT3 did not allow us to address this question without ambiguity (Payen et al., Molecular metabolism 33, 48, 2020).
Please rewrite the sentence on lines 441-443. The start of the sentence seems to say that the allele is null for lactate transport activity, but then the second part of the sentence seems like it contradicts the idea that there would be no lactate transport by this mutant protein.
We have modified the sentence as follows (lines 447-449): Our conclusion is that MCT3 with the rs77968014-G allele has a large deficit in lactate transport activity but it is more equivalent to an hypomorphic allele with a potential residual activity of the mutated allele expressed at very low level than to a real knock-out allele.
Also, if there is some residual MCT3 protein expressed shouldn't it be detected in the westerns and ICC?
We had indicated in the text (lines 441-443): The existence of a 209 aa domain as an intracellular loop would destabilize the mutant protein that would be degraded by the cell.
Round 2
Reviewer 3 Report
The authors satisfactorily addressed each of my original concerns.